# Evaluation of a Novel Immunochromatographic Device for Detecting *Porphyromonas gingivalis* in Patients with Periodontal Disease

**DOI:** 10.3390/ijms25158187

**Published:** 2024-07-26

**Authors:** Rieko Yamanaka, Michihiko Usui, Kaoru Kobayashi, Satoru Onizuka, Shingo Kasai, Kotaro Sano, Shou Hironaka, Ryota Yamasaki, Shinji Yoshii, Tsuyoshi Sato, Wataru Fujii, Masanori Iwasaki, Wataru Ariyoshi, Keisuke Nakashima, Tatsuji Nishihara

**Affiliations:** 1Division of Periodontology, Department of Oral Function, Kyushu Dental University, 2-6-1 Manazuru, Kokurakita-ku, Kitakyushu 803-8580, Fukuoka, Japan; r21yamanaka@fa.kyu-dent.ac.jp (R.Y.); r18onizuka@fa.kyu-dent.ac.jp (S.O.); r14kasai@fa.kyu-dent.ac.jp (S.K.); r14sano@fa.kyu-dent.ac.jp (K.S.); r22hironaka@fa.kyu-dent.ac.jp (S.H.); nakashimak@fa.kyu-dent.ac.jp (K.N.); 2Division of Infections and Molecular Biology, Department of Health Promotion, Kyushu Dental University, Kitakyushu 803-8580, Fukuoka, Japan; r20kobayashi@fa.kyu-dent.ac.jp (K.K.); r18yamasaki@fa.kyu-dent.ac.jp (R.Y.); arikichi@kyu-dent.ac.jp (W.A.); tatsujin@kyu-dent.ac.jp (T.N.); 3Division of Promoting Learning Design Education, Department of Physical Function, Kyushu Dental University, 2-6-1 Manazuru, Kokurakita-ku, Kitakyushu 803-8580, Fukuoka, Japan; r08yoshii@fa.kyu-dent.ac.jp; 4School of Oral Health Sciences, Faculty of Dentistry, Kyushu Dental University, 2-6-1 Manazuru, Kokurakita-ku, Kitakyushu 803-8580, Fukuoka, Japan; r23satou@fa.kyu-dent.ac.jp (T.S.); r15fujii@fa.kyu-dent.ac.jp (W.F.); 5Division of Preventive Dentistry, Department of Oral Health Science, Graduate School of Dental Medicine, Hokkaido University, Kita 13, Nishi 7, Kita-ku, Sapporo 060-8586, Hokkaido, Japan; iwasaki@den.hokudai.ac.jp

**Keywords:** immunochromatographic device, *Porphyromonas gingivalis*, periodontitis

## Abstract

*Porphyromonas gingivalis* is the most pathogenic periodontal bacterium in the world. Recently, *P. gingivalis* has been considered responsible for dysbiosis during the development of periodontitis. This study aimed to evaluate a novel immunochromatographic device using monoclonal antibodies against *P. gingivalis* in subgingival plaques. A total of 72 patients with chronic periodontitis and 53 periodontally healthy volunteers underwent clinical and microbiological examinations. Subgingival plaque samples were analyzed for the presence of *P. gingivalis* and compared using real-time polymerase chain reaction (PCR). In the periodontitis group, a significant positive correlation was observed between the test device scores and the real-time PCR results. The specificity, positive predictive value, negative predictive value, and accuracy of the test device for *P. gingivalis*, as determined by real-time PCR, were 98%, 94%, 89%, and 90%, respectively. There were significant differences in bacterial counts by real-time PCR among the groups with different ranges of device scores. Additionally, there was a significant positive correlation between the device scores for *P. gingivalis* and periodontal parameters. These results suggest that this novel immunochromatographic device can be effectively used for rapid detection and semi-quantification of *P. gingivalis* in subgingival plaques.

## 1. Introduction

Periodontal disease is an inflammatory disease in which the supporting tissues of the teeth are destroyed by a biofilm composed of periodontal bacteria. The progression of periodontal disease due to uncontrolled dental plaques containing periodontal bacteria results in alveolar bone resorption and tooth loss [1,2]. Microbiological assessment of periodontal bacteria in the periodontal pockets of patients with periodontal disease is important for evaluating and designing initial periodontal therapy and determining the need for periodontal surgery [3,4].

Currently, several bacterial tests are used in clinical practice to detect these periodontal bacteria. Testing systems using real-time polymerase chain reaction (PCR) are known to be perfectly accurate tests [5,6,7]. However, these tests are expensive and are not considered point-of-care tests. Enzymatic detection of trypsin-like enzyme activity in *Porphyromonas gingivalis, Tannerella forsythia,* and *Treponema denticola* is also used clinically at the chairside [8,9]. Compared with real-time PCR, these tests are less expensive and simpler to perform [5,6]. However, these three bacterial species are indistinguishable using this method. Therefore, there is a need for a system that can be used easily at the chairside and that can quickly and inexpensively detect specific periodontal bacteria.

*P. gingivalis*, a gram-negative anaerobic bacterium, is the major colonizer of subgingival pockets in patients with periodontitis and is involved in various forms of periodontitis [10,11]. *P. gingivalis*, together with *T. denticola* and *T. forsythia*, forms a “red complex” pathogen commonly associated with the destruction of periodontal tissue and the progression of periodontal disease [12,13]. *P. gingivalis* is known to have virulence factors including LPS, gingipain (a type of protease), and extracellular vesicles [14]. Recently, *P. gingivalis* has been implicated in the disruption of oral immune homeostasis, which is necessary for maintaining oral health, interfering with host immunity and enabling the emergence of dysbiotic communities [15,16]. Therefore, the detection and monitoring of *P. gingivalis* are clinically important for determining periodontal treatment plans and preventing the progression of periodontitis.

Recently, we developed an immunochromatographic device for detecting *P. gingivalis*. This immunochromatographic kit can detect cultured *P. gingivalis* strains within 15 min of sample application. We have also developed a reader that quantifies the detected bands and displays a score on a 12-point scale (0, 0.25, 0.5, 1, …, 4.5, 5.0) depending on the number of bacteria in culture (now submitting). However, it is unclear whether this device can detect *P. gingivalis* in clinical samples such as subgingival plaques. This study aimed to evaluate the detection and semi-quantitative performance of our novel immunochromatographic device for *P. gingivalis* in subgingival plaque samples and to compare it with the real-time PCR method.

## 2. Results

### Comparison of a Novel Immunochromatographic Device and Real-Time PCR Method in P. gingivalis Detection

We compared the device score of the novel immunochromatographic device with bacterial counts using the real-time PCR method to detect *P. gingivalis* in subgingival plaques from healthy subjects and patients with periodontitis. A statistically significant and strong positive correlation was observed between the two methods (r = 0.73, *p* < 0.05) (Figure 1). When tentative cutoff values of 0.25 in device score and 10^6^ colony forming unit/mL for the real-time PCR method were set, the specificity, positive predictive value, negative predictive value, and accuracy were 98%, 94%, 89%, and 90%, respectively. However, the sensitivity was slightly lower (77%; Table 1). The kappa coefficient of these tests was 0.78, which was higher than 0.6, indicating a substantial agreement. Next, we examined the distribution of *P. gingivalis* among the groups classified according to the score range of this device. Subgingival plaque samples were divided into three groups according to the test device scores: 0, 0.25–2.5, and 3.0–5.0. There were significant differences in the bacterial counts determined by real-time PCR among the groups (Figure 2), confirming the potential of this novel immunochromatographic device for semi-quantification.

The scores for this device in the subgingival plaques of patients with periodontal disease were significantly higher than those in the subgingival plaques of healthy individuals, which was similar to the real-time PCR results (Figure 3A,B). Next, the potential relationship between the detection of *P. gingivalis* using the two methods and clinical parameters was examined. There was a significant positive correlation between the test device score and periodontal parameters such as probing pocket depth (PPD) and clinical attachment level (CAL) at the sampled sites (Table 2). A significant positive correlation was observed among the bacterial counts of *P. gingivalis*, PPD, and CAL (Table 2).

To further clarify the significance of the score of the novel immunochromatographic device in assessing periodontal health, samples were divided into the following three groups according to the PPD and other conditions at the time of collection: 0 mm < PPD ≤ 3 mm, 4 mm ≤ PPD ≤ 6 mm, and 7 mm ≤ PPD. There were significant differences in device scores among the three groups (Figure 4A). The number of *P. gingivalis* in the samples determined using real-time PCR also showed comparable results to the novel immunochromatographic device score (Figure 4B).

Finally, a receiver operating characteristic (ROC) curve analysis was performed to examine the potential of the novel immunochromatographic kit and real-time PCR as diagnostic tests for periodontitis. The AUC between periodontitis and immunochromatographic device score was 0.73 (Figure 5A). In contrast, the AUC value between periodontitis and the number of *P. gingivalis* measured by real-time PCR was 0. 81 (Figure 5B).

## 3. Discussion

In this study, a novel device for the detection of *P. gingivalis*, PgS2303, based on a lateral flow immunochromatography method was evaluated using subgingival plaques from patients with periodontitis and healthy people. This immunochromatographic device was able to detect *P. gingivalis* in patient samples and showed high accuracy compared to real-time PCR. A positive correlation was also found between the device score and the count of *P. gingivalis* by real-time PCR. The novel immunochromatographic device, the PgS2303 device kit, enables quick identification (in approximately 15 min) of clinically relevant levels of *P. gingivalis*, providing results comparable to those of a reference real-time PCR method. 

As *P. gingivalis* is one of the most important pathogens in periodontal disease, understanding the presence of *P. gingivalis* in periodontal pockets is valuable for the treatment of periodontal disease. Various methods are used to detect periodontal bacteria. The real-time PCR method, which was compared with the immunochromatography method used in this study, can detect even a small number of *P. gingivalis* by amplifying their DNA, and is highly quantitative. However, the equipment used for the PCR method is large and the cost of detection is high. Additionally, real-time PCR cannot detect *P. gingivalis* in a short period. Our previously developed kit for detecting the BANA-degrading enzymes produced by red complex pathogens, including *P. gingivalis*, can detect these enzymes quickly and easily. However, it cannot specifically detect *P. gingivalis*, because *T. forsythia* and *T. denticola* also produce the same enzymes. The novel immunochromatography kit validated in this study can detect the presence of *P. gingivalis* in approximately 15 min. The reader device that detects the colored bands in the immunochromatography kit is also small and can be placed at the patient’s chairside. The ability to determine the presence or absence of *P. gingivalis* in a patient’s periodontal pocket on the same day is beneficial for both the patient and dentist, as a future treatment plan can be decided upon immediately.

For the detection of *P. gingivalis*, a positive correlation was observed between the immunochromatographic score and the number of *P. gingivalis* bacteria as evaluated using real-time PCR (r = 0.73, Spearman’s rank correlation coefficient). When the distribution of *P. gingivalis* was evaluated among groups classified according to the score range of the immunochromatographic kit, significant differences were found in the number of bacteria (evaluated by real-time PCR). These results indicate the potential for semi-quantification of *P. gingivalis* in clinical samples using this kit, and the relationship between *P. gingivalis* detection and periodontal tissue destruction is well documented [17]. The number of *P. gingivalis* in subgingival biofilms increases in areas with higher PPD and bleeding on probing (BOP) [18]. After scaling and root planning, there were areas with lower PPD reduction and BOP [19]. Some researchers have shown a significant positive correlation between the detection of *P. gingivalis* using real-time PCR and periodontal parameters such as PPD and CAL [20]. In the present study, we found a significant positive correlation between the detection of *P. gingivalis* by real-time PCR, PPD, and CAL. The scores obtained using the novel immunochromatographic kit also showed significant positive correlations with PPD and CAL. These findings suggest the potential clinical application of this novel device.

In the present study, we used real-time PCR and a novel immunochromatography kit to confirm the presence of *P. gingivalis* in the deepest pockets of patients with periodontal disease. Surprisingly, the detection rate was 70% using real-time PCR, which also measures dead bacteria. The detection rate of the new immunochromatographic kit was 60%. These data supported the hypothesis that periodontitis is an inflammatory disease caused by various periodontal bacterial infections. In other words, *P. gingivalis* is not always present in the deep periodontal pockets where periodontal disease is diagnosed. The idea of using immunochromatography to screen for periodontal disease has emerged because it is an easy and inexpensive method for detecting periodontal bacteria. However, it is not practical to use an immunochromatographic kit for screening for *P. gingivalis* in periodontitis. It is necessary to comprehensively determine the bacteria involved in the pathogenesis of periodontitis, such as *T. forsythia* and *T. denticola*, which are components of the red complex, as well as other periodontopathic bacteria such as *Fusobacterium nucleatum* and *Aggregatibacter actinomycetemcomitans*. In contrast, *P. gingivalis* has been detected in periodontal pockets as deep as 3 mm, which are considered clinically healthy, although the percentage was quite small. It may be significant that anaerobic bacteria such as *P. gingivalis* are detected in such shallow pockets. This is because *P. gingivalis* is considered a keystone bacterium. By disrupting the host’s innate immune system and impairing leukocyte function, it not only promotes the survival and expansion of *P. gingivalis* itself, but also facilitates the survival and growth of other bacterial communities. In other words, the shallow pockets may deepen in the future. Therefore, shallow periodontal pockets where *P. gingivalis* is present may require careful monitoring using immunochromatographic kits. 

Periodontal treatment consists of active periodontal therapy, including scaling, root planing, periodontal surgery, and maintenance or supportive periodontal therapy. Ideally, *P. gingivalis* should be mechanically removed from periodontal pockets by active periodontal therapy and should not be present during maintenance or supportive periodontal therapy. If periodontal pocket depths do not decrease and *P. gingivalis* is still detected despite active periodontal therapy, antimicrobial therapy should be considered. Furthermore, if *P. gingivalis* is detected using this immunochromatographic device during maintenance or supportive periodontal therapy, the interval between treatments may need to be shortened to prevent the progression of periodontal disease. 

In this study, the specificity of the novel immunochromatographic device for the real-time PCR method was extremely high at 98% when tentative cutoff values of 0.25 in device score and 10^6^ colony forming unit/mL for the real-time PCR method were set, whereas the sensitivity was not as high, at 77%. However, in preliminary experiments, this immunochromatographic device was able to detect cultured strains of *P. gingivalis* JCM 12257 at 10^4^ colony-forming unit/mL. This difference in the detection limit of the device may be due to differences between samples from cultured bacterial strains and clinical subgingival plaques. Subgingival plaque from patients also contains biofilms consisting of bacteria other than *P. gingivalis* and extrabacterial polysaccharides; therefore, it may be difficult to degrade them and detect antigens of *P. gingivalis*. In this immunochromatography kit, TritonX-100 was used as the sample extraction buffer. To solve this problem, it may be necessary to examine the conditions of the extraction solution in more detail, such as by adjusting the concentration and pH of TritonX-100 or, in some cases, using an extraction solution other than TritonX-100.

Over the past several decades, the relationship between periodontal disease and systemic diseases has been studied. Inflammation resulting from periodontal disease and periodontal bacteria themselves can impact various diseases, including diabetes [21,22,23]. Recently, using a periodontal bacterial enzyme detection kit, we reported that high trypsin-like enzyme activity produced by red complex species, including *P. gingivalis*, in samples collected from the tongue, was associated with decreased renal function [24]. Several investigators have also reported the involvement of *P. gingivalis* in the relationship between periodontal disease and Alzheimer’s disease [25,26,27]. Furthermore, it has been reported that *P. gingivalis* is implicated in the etiology of rheumatoid arthritis, an autoimmune disease [28,29,30]. Thus, since the presence of *P. gingivalis* is associated with periodontitis as well as other diseases of the body, it is very important to monitor the oral cavity, the home of *P. gingivalis*, using periodontal bacterial enzyme detection kits and immunochromatography kits, which are inexpensive and easy to use. 

This study has some limitations. It was not possible to match the age and sex of the groups. However, whether this affects the results remains unknown. In future, we plan to adjust for these conditions and investigate the usefulness of this device in the evaluation of periodontal treatment.

## 4. Materials and Methods

### 4.1. Study Population

Participants were consecutively recruited from patients with periodontitis or healthy volunteers who visited Kyushu Dental University Hospital or Steel Memorial Yahata Hospital in Kitakyushu City, Fukuoka, Japan. Written informed consent was obtained from all participants. The participants received no medication that could affect their periodontal condition, such as antibiotics or anti-inflammatory drugs, for at least three months. According to the new 2018 classification of periodontitis, patients with ≥2 interproximal sites with a CAL ≥ 2 mm (not on the same tooth and non-adjacent teeth) or ≥2 buccal lingual sites with a PPD of ≥3 mm (not on the same tooth) with ≥3 mm attachment loss were diagnosed with periodontitis [31]. Periodontitis was diagnosed by the periodontists (M. U.). This study was conducted in accordance with the principles of the Declaration of Helsinki and approved by the Ethics Committee of Kyushu Dental University (approval number: 20-68). G*power (v3.1.9.6) software (https://www.psychologie.hhu.de/) was used to calculate the sample size, with a mean effect size (Cohen’s D) of 0.9 and a power of 80%. Forty subjects (40 periodontal pockets) were required in each group (periodontitis and healthy subjects) to demonstrate differences between the groups (Table 3).

### 4.2. Subgingival Plaque Sampling

Subgingival plaque samples were obtained from the deepest PPD in patients with periodontitis and in healthy individuals. The teeth were air-dried, and the supragingival plaque was removed and then isolated with cotton rolls. Subgingival plaque samples were collected by inserting two sterile paper points (#40, Yoshida Co., Tokyo, Japan) into the deepest area of the accessible pocket for 30 s. They were immediately transferred into a sampling tube supplied in the immunochromatographic device kit and were stored at −80 °C. Before use, the samples were thawed and mixed thoroughly in a vortex mixer.

### 4.3. Immunochromatographic Device for P. gingivalis

Immunochromatography kits for *P. gingivalis* PgS2303 consisted of a test cassette and a sample tube with sample extraction buffer. The membrane was coated with a specific monoclonal antibody against *P. gingivalis* (catching antibody) at 2 mg/mL. The sample extraction buffer was diluted 10-fold with 1 × PBS (-) in purified water and TritonX-100 (Sigma-Aldrich, St. Louis, MO, USA) was added to reach a concentration of 1%. A volume of 100 µL of the sample dissolved in sample extraction buffer was dispensed into the sample drop section of the test cassette, and the band was observed after 15 min. Colored bands were evaluated using a measurement reader. (Figure 6) The intensity of band coloration was evaluated using a 12-point scoring system: 0, 0.25, 0.5, 1.0, 1.5, 2.0, 2.5, 3.0, 3.5, 4.0, 4.5, and 5.0. The score for the coloration intensity of each band was determined using cultured *P. gingivalis* JCM 12257.

### 4.4. Real-Time PCR Method

Real-time PCR was used to determine the number of *P. gingivalis* cells in each sample. Genomic DNA was isolated from subgingival plaque samples lysed in extraction buffer using the QIAamp DNA Mini Kit (Qiagen, Hilden, Germany) according to the manufacturer’s recommendations. A StepOne RT-PCR System (Applied Biosystems, Foster City, CA, USA) was used to amplify and detect DNA with *P. gingivalis*-specific primers. For each real-time PCR, universal SYBR Green Supermix (Bio-Rad Laboratories, Hercules, CA, USA) and a total PCR amplification volume of 20 μL for each reaction were used. The DNA amplification conditions for PCR with species-specific primers for *P. gingivalis* were as follows: 30 s initial denaturation at 95 °C, followed by 50 consecutive cycles at 95 °C for 10 s, 60 °C for 20 s, and 72 °C for 20 s, for data collection. The number of *P. gingivalis* samples was calculated by using a calibration curve. The forward and reverse primer sequences for *P. gingivalis* were as follows: F, CCGCATACACTTGTATTATTGCATGATATT; R, AAGAAGTTTACAATCCTTAGGACTGTCT. The primer sets used were obtained from Sakamoto et al. [32]. Analysis of clinical samples using real-time PCR and immunochromatographic devices was performed in a blinded fashion.

### 4.5. Statistical Analysis

Differences in demographic and clinical parameters between the groups were assessed using the Mann–Whitney U test. Differences in Spearman’s rank correlation were used to analyze the correlation between the results of the immunochromatographic device and the real-time PCR method. Spearman’s rank correlation was used to determine the correlation between the detection results and clinical periodontal parameters. Differences in *P. gingivalis* numbers and device scores among the different device score groups were determined using the Steel-Dwass method. Statistical significance was set at *p* < 0.05. 

## 5. Conclusions

In this study, the novel immunochromatographic device, PgS2303, was able to detect *P. gingivalis* quickly and with high accuracy. *P. gingivalis* is closely associated with systemic diseases. Therefore, monitoring of *P. gingivalis* in the oral cavity periodically using such a kit may help improve systemic health.

## Figures and Tables

**Figure 1 ijms-25-08187-f001:**
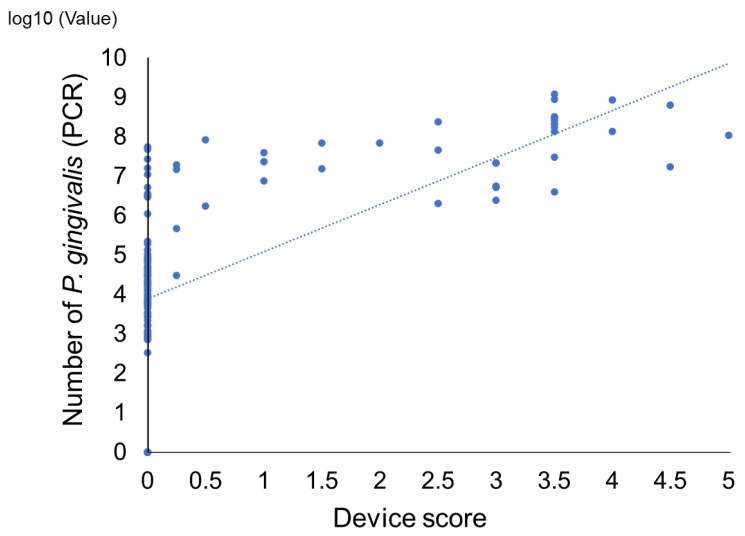
Correlation between score of the novel immunochromatographic device and number of *P. gingivalis* by real-time PCR in subgingival plaque. A significant positive correlation was found between these methods (correlation coefficient: 0.73. *p* < 0.005).

**Figure 2 ijms-25-08187-f002:**
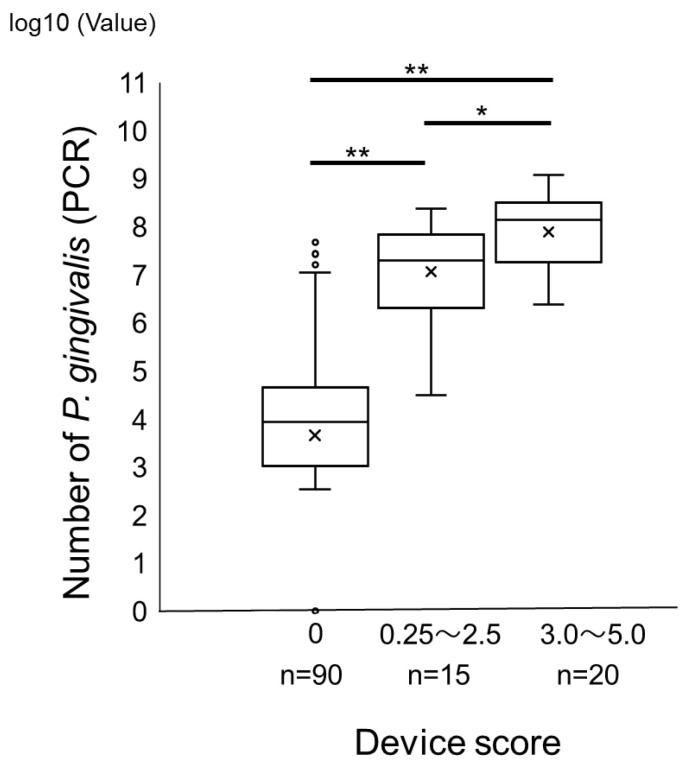
Distribution of the number of *P. gingivalis* among groups classified by the score ranges of subgingival plaque. The subgingival plaque samples were divided according to the device score into groups (0, 0.25–2.5 and 3.0–5.0). * *p* < 0.05, ** *p* < 0.001. × signs represent the mean of values in groups. Circles indicate the outlier.2.2. Relationship between the Score of the Novel Immunochromatographic Device and Periodontal Clinical Parameters.

**Figure 3 ijms-25-08187-f003:**
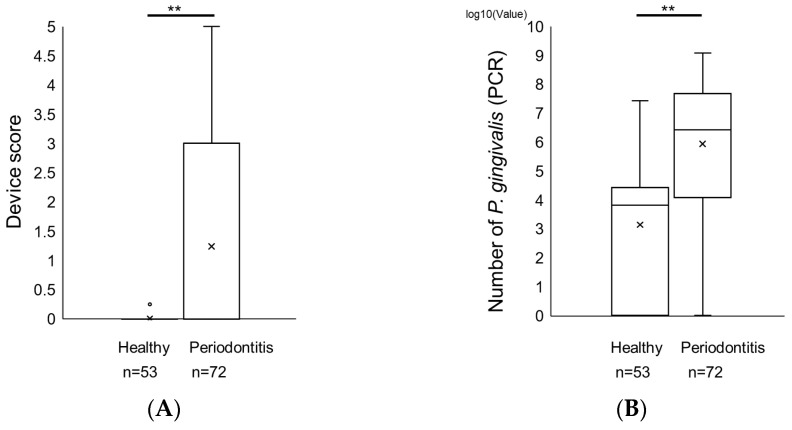
Comparison of score of the novel immunochromatographic device and number of *P. gingivalis* by real-time PCR in subgingival plaque from patients with periodontitis and healthy individuals. (**A**) Score of the device assessed by the test device and (**B**) the number of *P. gingivalis* measured by real-time PCR in subgingival plaque in patients with periodontal disease and healthy subjects. ** *p* < 0.001. × signs represent the mean of values in groups. Circles indicate the outlier.

**Figure 4 ijms-25-08187-f004:**
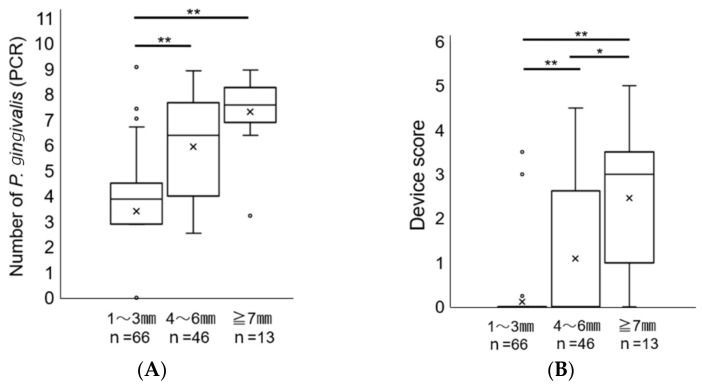
Comparison of score of the novel immunochromatographic device and number of *P. gingivalis* by real-time PCR at different periodontal pocket depths. (**A**) Score assessed by the novel immunochromatographic device and (**B**) number of *P. gingivalis* measured by real-time PCR. ** *p* < 0.001; * *p* < 0.05. × signs represent the mean of values in groups. Circles indicate the outlier.

**Figure 5 ijms-25-08187-f005:**
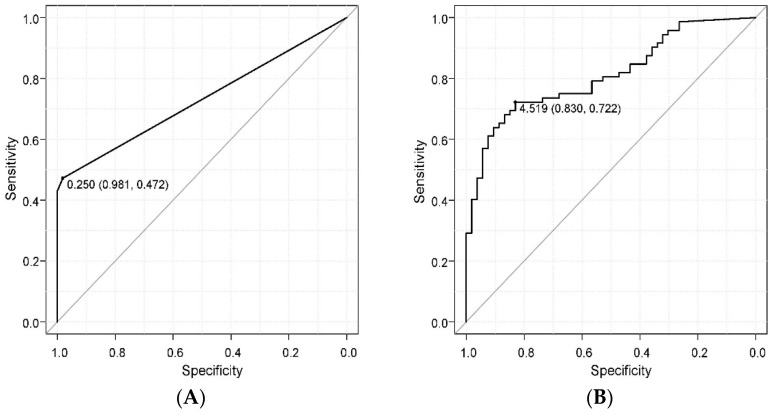
Evaluation of diagnostic capability for periodontitis using the novel chromatographic device and real-time PCR with ROC curves. (**A**) Score of the novel chromatographic device. AUC: 0.73, 95% confidence interval: 0.671–0.791, cutoff value: 0.25. (**B**) The number of *P. gingivalis* measured by real-time PCR. AUC: 0.81, 95% confidence interval: 0.737–0.885, cutoff value: 0.459.

**Figure 6 ijms-25-08187-f006:**
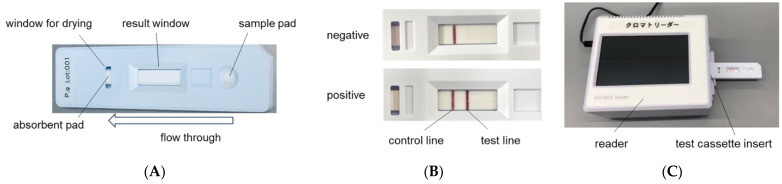
Schematic of the novel immunochromatographic device for *P. gingivalis*, PgS2303. (**A**) Test cassette of immunochromatographic device for *P. gingivalis.* (**B**) A test line appears alongside the control line after 15 min when the sample contains *P. gingivalis*. (**C**) Test lines are detected and scored by the reader.

**Table 1 ijms-25-08187-t001:** Comparison of the novel immunochromatographic device and real-time PCR (real-time PCR) in subgingival plaque samples.

Comparison of the Novel Immunochromatographic Device and Real-Time PCR (Real-Time PCR) in Subgingival Plaque Samples									
		real-time PCR			sensitivity	specificity	PPV	NPV	accuracy
		POS	NEG	TOTAL	77%	98%	94%	89%	90%
TEST DEVICE	POS	33	2	35					
	NEG	10	80	90					
	TOTAL	43	82	88					
X^2^ (*p* value)		*p* < 0.01							

POS, positive; NEG, negative; PPV, positive predictive value; NPV, negative predictive value. *p* < 0.01 (chi-squared test). The kappa coefficient was 0.78.

**Table 2 ijms-25-08187-t002:** Correlation between the detection of *P. gingivalis* and periodontal clinical parameters.

Parameters	Device Score (Test Device)	Number of *P. gingivalis* (Real-Time PCR)
PPD (mm)	0.612	0.594
CAL (mm)	0.605	0.606
BOP	0.403	0.365

Spearman’s rank correlation coefficient (r). All values indicated a statistically significant correlation (*p* < 0.01). PPD, probing pocket depth; CAL, clinical attachment level; BOP, bleeding on probing.

**Table 3 ijms-25-08187-t003:** Characteristics of study subjects and periodontal parameters at sampled site.

	Healthy Subjects	Patients with Periodontitis	Difference
Number Male Female	n = 53 23 30	n = 72 36 36	
Age (average)	47.51 ± 21.10	69.81 ± 11.82	*p* < 0.01
PPD (mm)	2.32 ± 0.51	4.96 ± 1.54	*p* < 0.01
CAL (mm)	0.74 ± 1.07	5.60 ± 1.97	*p* < 0.01
BOP	0.06 ± 0.23	0.40 ± 0.49	*p* < 0.01

PPD, probing pocket depth; CAL, clinical attachment level; BOP, bleeding on probing. Data are shown as the mean ± standard deviation. The average BOP was calculated as follows; BOP+ = 1, BOP− = 0.

## Data Availability

Data supporting the findings of this study are available from the corresponding author, M.U., upon reasonable request.

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
