# Peer review of "Evaluation of a Novel Immunochromatographic Device for Detecting Porphyromonas gingivalis in Patients with Periodontal Disease"

_ijms, 2024, doi:10.3390/ijms25158187_

Round 1

Reviewer 1 Report

Comments and Suggestions for Authors

I consider this work to be useful, and the conclusion clearly indicates its potential use in populations of individuals with systemic diseases that have potentially been related to Porphyromonas gingivalis. However, the study reports a sensitivity of 77% and a specificity of 98%. In my understanding, this performance relationship is not ideal for a screening test, where I would expect the opposite relationship and would sacrifice specificity. I believe this point needs to be at least mentioned and discussed, including some improvement plans, with the aim of achieving a useful tool in the scenario described in the conclusion. There is no mention of a therapeutic decision algorithm based on the test result. I consider this important, even though in dental practice antimicrobial treatment is not standardized in a system to reduce their use, it might also be interesting to explore these contexts. Nevertheless, I think it is a useful, concrete work. For it to have a real impact on individuals with periodontitis or other systemic diseases in the short to medium term, there should indeed be a plan to improve sensitivity.

Please replace some of the self-citations as there are several. I would look for another way to highlight in the paper that this is a robust group in this area of research

Author Response

We thank you for your accurate comments and suggestions on our manuscript. In accordance with your suggestion, we have revised our manuscript as follows

Comment 1.

 In my understanding, this performance relationship is not ideal for a screening test, where I would expect the opposite relationship and would sacrifice specificity. I believe this point needs to be at least mentioned and discussed, including some improvement plans, with the aim of achieving a useful tool in the scenario described in the conclusion.

As you pointed out, the sensitivity of the novel immunochromatographic device was 77%, which is lower than we had expected. In the discussion part, we discussed the reasons for this and how to solve the problem (P7 L 225-238).

Comment 2.

 There is no mention of a therapeutic decision algorithm based on the test result. I consider this important, even though in dental practice antimicrobial treatment is not standardized in a system to reduce their use, it might also be interesting to explore these contexts. 

Thank you for your very perceptive and thought-provoking points. It is clinically very important to clarify what periodontal treatment is to be performed for P. gingivalis detected by this immunochromatographic device. We have described these points including antimicrobial therapy in the Discussion Part (P7 L 216-224).

Comment 3.

Please replace some of the self-citations as there are several.

We have made some changes to several cited papers in this manuscript. Thank you for pointing this out.

Reviewer 2 Report

Comments and Suggestions for Authors

The manuscript entitled “Evaluation of a novel immunochromatographic device for detecting Porphyromonas gingivalis in patients with periodontal disease” is a clinical study to evaluate an alternative to detecting P. gingivalis DNA in dental plaque. The detection is semi-quantitative, relying on the color of the band, using a band that is similar to a covid auto test. The study validates the methodology in the deepest periodontal pocket of patients. In general, the manuscript is quite good, with only minor modifications needed.

Introduction:

Authors point out the relevance of periodontitis, the bacteria involved and the advantages and limitations of current systems to detect periopathogens. The objectives are clear, the structure is adequate, and it is overall well written.

Methods:

Authors describe the informed consent and the criteria to diagnose periodontitis. Moreover, ethical approval and sample size a priori were included.

The pictures of the band and the device and useful and illustrate the proposed methodology.

The only suggestion is to add information about the “blindness” of the evaluators regarding the data analysis, which is relevant for this type of study.

Results:

In general, results are clear and easy to interpret.

Line 90: 10^6. Check the formatting.

Table 1: the structure of the table seems confusing. The reviewer struggled to understand the device and PCR results. The percentages are seamless, however, the first part of the table (on the left), which contains the positive and negative numbers is not obvious. Thus, the reviewer suggests a reformulation of this part.

Discussion:

Authors question the utility in detecting only P. gingivalis in comparison to the previous system that unspecific detects red complex bacteria. The reviewer agrees with the authors. Being all three bacteria Gram-negative and anaerobe, the therapeutics would be similar, with or without P. gingivalis. However, by detecting P. gingivalis in (still) healthy sites can be particularly useful, directing the prevention in these specific sites. Authors pointed out the difference in age between groups as a limitation.

Conclusion:

Adequate.

The manuscript entitled “Evaluation of a novel immunochromatographic device for detecting Porphyromonas gingivalis in patients with periodontal disease” is a clinical study to evaluate an alternative to detecting P. gingivalis DNA in dental plaque. The detection is semi-quantitative, relying on the color of the band, using a band that is similar to a covid auto test. The study validates the methodology in the deepest periodontal pocket of patients. In general, the manuscript is quite good, with only minor modifications needed.

Introduction:

Authors point out the relevance of periodontitis, the bacteria involved and the advantages and limitations of current systems to detect periopathogens. The objectives are clear, the structure is adequate, and it is overall well written.

Methods:

Authors describe the informed consent and the criteria to diagnose periodontitis. Moreover, ethical approval and sample size a priori were included.

The pictures of the band and the device and useful and illustrate the proposed methodology.

The only suggestion is to add information about the “blindness” of the evaluators regarding the data analysis, which is relevant for this type of study.

Results:

In general, results are clear and easy to interpret.

Line 90: 10^6. Check the formatting.

Table 1: the structure of the table seems confusing. The reviewer struggled to understand the device and PCR results. The percentages are seamless, however, the first part of the table (on the left), which contains the positive and negative numbers is not obvious. Thus, the reviewer suggests a reformulation of this part.

Discussion:

Authors question the utility in detecting only P. gingivalis in comparison to the previous system that unspecific detects red complex bacteria. The reviewer agrees with the authors. Being all three bacteria Gram-negative and anaerobe, the therapeutics would be similar, with or without P. gingivalis. However, by detecting P. gingivalis in (still) healthy sites can be particularly useful, directing the prevention in these specific sites. Authors pointed out the difference in age between groups as a limitation.

Conclusion:

Adequate.

Comments on the Quality of English Language

Minor editing of English language required

Author Response

Thank you for your very kind and thoughtful review.

Comment 1.

 Methods

 The only suggestion is to add information about the “blindness” of the evaluators regarding the data analysis, which is relevant for this type of study.

Biochemical analysis of clinical samples using real-time PCR and immunochromatographic devices was performed in a blinded fashion in this study. This information has been added to Materials and Methods (P9L317-318).

Comment 2.

Results

Table 1: the structure of the table seems confusing. The reviewer struggled to understand the device and PCR results. The percentages are seamless, however, the first part of the table (on the left), which contains the positive and negative numbers is not obvious. Thus, the reviewer suggests a reformulation of this part.

Thank you for your useful comment.

We have reconstructed Table1 according to your suggestion.